# Dynamic Alterations to Hepatic MicroRNA-29a in Response to Long-Term High-Fat Diet and EtOH Feeding

**DOI:** 10.3390/ijms241914564

**Published:** 2023-09-26

**Authors:** Tiebing Liang, Janaiah Kota, Kent E. Williams, Romil Saxena, Samer Gawrieh, Xiaoling Zhong, Teresa A. Zimmers, Naga Chalasani

**Affiliations:** 1Division of Gastroenterology and Hepatology, Department of Medicine, Indiana University School of Medicine, Indianapolis, IN 46202, USA; kew5@iu.edu (K.E.W.); sgawrieh@iu.edu (S.G.); nchalasa@iu.edu (N.C.); 2Ultragenyx Pharmaceuticals, Novato, CA 94949, USA; kotajanaiah@gmail.com; 3Department of Pathology and Laboratory Medicine, Indiana University School of Medicine, Indianapolis, IN 46202, USA; rsaxena@iu.edu; 4Department of Surgery, Indiana University School of Medicine, Indianapolis, IN 46202, USA; xzhong@iu.edu (X.Z.); zimmerst@iu.edu (T.A.Z.); 5Indiana Center for Musculoskeletal Health, Indianapolis, IN 46202, USA; 6Richard L. Roudebush Veterans Administration Medical Center, Indianapolis, IN 46202, USA

**Keywords:** high-fat diet, EtOH, liver, microRNA, long-term feeding

## Abstract

MicroRNA-29a (miR-29a) is a well characterized fibro-inflammatory molecule and its aberrant expression is linked to a variety of pathological liver conditions. The long-term effects of a high-fat diet (HFD) in combination with different levels of EtOH consumption on miR-29a expression and liver pathobiology are unknown. Mice at 8 weeks of age were divided into five groups (calorie-matched diet plus water (CMD) as a control group, HFD plus water (HFD) as a liver disease group, HFD plus 2% EtOH (HFD + 2% E), HFD + 10% E, and HFD + 20% E as intervention groups) and fed for 4, 13, 26, or 39 weeks. At each time point, analyses were performed for liver weight/body weight (BW) ratio, AST/ALT ratio, as well as liver histology assessments, which included inflammation, estimated fat deposition, lipid area, and fibrosis. Hepatic miR-29a was measured and correlations with phenotypic traits were determined. Four-week feeding produced no differences between the groups on all collected phenotypic traits or miR-29a expression, while significant effects were observed after 13 weeks, with EtOH concentration-specific induction of miR-29a. A turning point for most of the collected traits was apparent at 26 weeks, and miR-29a was significantly down-regulated with increasing liver injury. Overall, miR-29a up-regulation was associated with a lower liver/BW ratio, fat deposition, inflammation, and fibrosis, suggesting a protective role of miR-29a against liver disease progression. A HFD plus increasing concentrations of EtOH produces progressive adverse effects on the liver, with no evidence of beneficial effects of low-dose EtOH consumption. Moreover, miR-29a up-regulation is associated with less severe liver injury.

## 1. Introduction

The consumption of a Western-style high-fat diet (HFD) and alcohol has recently become typical for many individuals across the globe. The prevalence of obesity and excessive alcohol consumption are rising and with them come increased risks of liver diseases and incidences of mortality, along with greater societal expense [1,2,3,4,5]. The resulting clinical consequences are likely to become more prevalent in the coming decades, as diet-induced liver disease has the potential to affect both the quality and quantity of life, and poses transgenerational risks as well [6,7,8]. However, investigation of the combinatorial effect of a high-fat diet and EtOH consumption is limited, and studies on long-term feeding are lacking. The primary drawback of human studies is that the information gained is often based solely on self-reporting, making it difficult to accurately determine the drinking level. Longitudinal human studies are few and present tremendous challenges [9,10,11,12]. Animal models have become the primary tool for research, as measures of ethanol consumption in animal studies are generally more precise than those reported in human studies [13]. Both large and small animal models have been used to test dietary effects on metabolic syndrome [14,15,16]. 

The mechanisms by which alcohol [17,18,19] or HFD consumption individually contribute to liver damage have been previously demonstrated [16,20,21]. There is a consensus that the modern Western diet produces significant negative effects on health, contributing to an increase in the prevalence of NAFLD. Regarding alcohol consumption, the opinions are more varied. Some studies have found light-to-moderate alcohol consumption may be beneficial for heart disease, diabetes, or nonalcoholic fatty liver disease (NAFLD), but such findings have not been demonstrated universally [22,23,24]. Interestingly, a human epidemiological study found that compared with lifelong abstainers, individuals who consumed <5, <15, and <30 g EtOH per day exhibited adjusted relative risks of developing T2DM of 0.8, 0.67, and 0.42, respectively [25]. The alcohol intake amount has a dose-dependent role in conferring protection from T2DM. Other research suggests that binge alcohol intake combined with obesity can promote liver injury, steatohepatitis, and fibrosis. Thus, studying the combinatorial effect of alcohol and a HFD is relevant for understanding disease development [26,27,28].

Contradictory results have been found in both human and animal research into metabolism and alcohol intake. Moderate alcohol intake revealed a negative association with steatohepatitis and was positively associated with hepatocellular carcinoma in NAFLD [29]. Others reported that moderate ethanol intake and HFD feeding for 3 months was beneficial for reducing NAFLD, and that chronic moderate alcohol intake can ameliorate high-fat, high-cholesterol-induced liver fibrosis, but not hepatic inflammation [23]. More studies found that alcohol consumption relieves or aggravates HFD-induced steatohepatitis [30,31,32]. Even though many individuals consume alcohol from young adulthood until old age, the effects of prolonged alcohol and high-fat diet consumption were not adequately addressed in these previous studies. Additionally, interactions between a HFD and EtOH intake have been shown to affect the microbiome, skin wound healing, and the inflammatory response, highlighting the potential of dietary intervention as an approach for preventing the progression of disease more generally [6,33,34,35].

Recently, microRNA (miRNA) regulation has become a novel therapeutic strategy to treat liver disease [36,37,38], and aberrant expression of miRNAs has been observed to result in alterations in essential physiological processes in numerous pathological conditions, including those of the liver [39,40,41]. The miR-29 family consists of three members (i.e., 29a, -29b, and -29c) encoded by two polycistronic miRNA clusters, miR-29a/b1 and miR-29b2/c. Among the three family members, miR-29a is the most abundantly expressed in the liver and other organs [42,43]. A body of evidence has demonstrated that miR-29a functions as an anti-fibrotic and anti-inflammatory molecule in various organs, including the liver [44,45]. A TGF-β1-mediated down-regulation of miR-29a was observed in several fibrotic processes, including diseases of the liver, lung, heart, and kidney, highlighting miR-29a as an important target deserving further investigation [43,46,47,48,49].

Previous studies have suggested that energy compensation in response to beverage consumption is generally negligible, and no alterations in food intake were found when EtOH was provided with a HFD [50]. Thus, food intake was not assessed in this study. Our study was designed to understand the effect of a combined HFD and EtOH on the liver and associated physiological changes with long-term feeding, and to understand alterations in hepatic miR-29a expression and its correlation with collected phenotypes.

### New and Noteworthy

Modeling human high-fat diet (HFD) and ethanol consumption from adolescence to old age.The short- to long-term feeding design (4 weeks, 13 weeks, 26 weeks, and 39 weeks), multiple feeding groups (CMD, HFD, HFD + 2%, 10%, or 20% *v*/*v* E), and data collection at multiple time points are the strengths of this study.Short-term feeding in 8-week-old mice demonstrated no significant changes, but longer feeding lengths resulted in significant physiological changes, including liver fibrosis.HFD feeding with chronic consumption of a high concentration of EtOH (20% E) was able to counteract weight gain, but increased mortality; a HFD with chronic consumption of a low concentration of EtOH (2% E) increased liver fibrosis.A HFD plus EtOH feeding drove significant initial up-regulation of miR-29a after 13 weeks but resulted in the down-regulation of miR-29a in association with liver injury after 26 weeks.

## 2. Results

### 2.1. Long-Term Feeding of 20% EtOH Counters HFD-Induced Body Weight Gain

EtOH and water consumption over time were examined (Appendix A). No differences were observed in the average daily liquid consumption (Appendix A). The average EtOH intake (g EtOH/kg BW/day) for the four treatment lengths were calculated (Appendix A). As expected, the EtOH intake of each group was significantly different from the other groups (*p* < 0.001) and enabled the modeling of different levels of voluntary EtOH drinking behavior in humans (Appendix A). Mice at ages corresponding to human adolescence drank the highest levels, and both EtOH and water consumption decreased with age, with a slight increase after 30 weeks of feeding.

The growth curves on the average BW of all the available animals for each week are shown in Figure 1B. While all group weights clustered together during the first 4 weeks, the HFD plus 20% E group diverged significantly from the CMD by week 13. Interestingly, a different trend in body weight change was observed at around 15–17 weeks, with all HFD-containing groups gaining less weight than the CMD (Figure 1A, gray box area). Between 13 and 26 weeks, the CMD group overtook the HFD group as having the highest average body weight and began to pull away from the other four groups. By week 32, the average body weight of the CMD group was significantly higher than the other four groups. The peak BW was evident in all groups by week 26, a time frame corresponding to humans in their 30s to 40s, with the HFD + 20% E group diverging significantly from the other groups. Trends in decreasing weight were observed after 26 weeks in all groups, which corresponds to humans in their 40–50s (Figure 1A,B). The percentage changes in BW relative to the starting weight were significantly different from the CMD at 39 weeks (Appendix A). Overall, we found that animals from the HFD plus 20% E feeding group exhibited significantly less BW gain than the HFD alone, with a dosage-dependent response trend only after long-term feeding. 

### 2.2. Dynamic Alteration of miR-29a Expression in Response to Feeding

Hepatic miR-29a expression was measured to observe its response to the combination of a HFD and EtOH. Although hepatic miR-29a expression was similar in all groups after 4 weeks of feeding, an EtOH concentration-specific response became evident after 13 weeks (Figure 2A). At 13 weeks, an increase in miR-29a expression was observed in all HFD + E groups, and a significant increase in miR-29a was observed in the HFD + 10% E (*p* < 0.0001) and 20% E (*p* < 0.001) groups. All HFD + E groups peaked in miR-29a expression at 13 weeks of feeding, but the CMD and HFD peaked after 26 weeks of feeding, with the CMD exhibiting significantly higher levels than the HFD + E groups (*p* < 0.01) (Figure 2A,B). After this time point, miR-29a expression declined in all groups and no significant differences were observed between the groups. Thus, the HFD + E groups have an earlier induction of miR-29a expression than non-EtOH groups.

The correlation of miR-29a expression and the collected phenotypes, including the BW, AST, ALT, AST/ALT ratio, liver weight, liver/BW ratio, lipid area, and epididymal fat, were performed at each time point. No correlation was found between miR-29a expression and all the collected phenotypes at the 4-week time point. Few correlations between miR-29a expression and the phenotypes were found, which included a positive correlation with the AST/ALT ratio at 13 and 26 weeks, and a negative correlation with the lipid area at 26 and 39 weeks. The data suggest that hepatic miR-29a expression responds to EtOH in a concentration-specific manner and is feeding length dependent, which reflects the involvement of miR-29a in the combinatorial effect of a HFD and EtOH on liver disease progression.

### 2.3. Feeding Length Results on Liver and Epididymal Fat Mass Alterations

To understand the systematic changes that occur during feeding, the effect of a CMD, HFD, and EtOH on organ weights, including liver and epididymal fat, were measured. Liver weights (Appendix A) and average liver weight:BW ratios (Figure 3A) were determined. The CMD had the lowest liver weight gain and liver weight:BW ratio, and was significantly lower than all the HFD-containing groups at 26 and 39 weeks (Figure 3A). We next looked at the correlation between liver weight and miR-29a expression. Due to the limited sample size of the feeding groups at each time point, we opted to compare the mice representing the 25% highest and 25% lowest quartiles of miR-29a expression. Lower miR-29a expression was associated with a trend toward a higher liver weight and a significantly higher liver weight:BW ratio (*p* < 0.05) (Figure 3B). Subsequent analysis of all the samples confirmed a negative correlation between miR-29a expression and the liver weight:BW ratio (*p* < 0.05) (Figure 3C).

Alcohol, adipose tissue abundance, and liver disease development are interconnected [26]. The effects of a HFD and EtOH consumption on epididymal fat weight (Appendix A), and the epididymal fat weight:BW ratio (Figure 3D) were also analyzed. While the weights and ratios in all the groups rose from 4 to 26 weeks, the CMD plateaued after week 13. By week 26, all HFD-containing groups had significantly higher epididymal fat weight and epididymal fat:BW ratios than the CMD group. By week 39, the HFD and HFD + 2% E had a significantly higher epididymal fat weight and epididymal fat:BW ratios than the CMD group. However, no correlation between the hepatic miR-29a with the epididymal fat weight or the epididymal fat:BW ratio was found. Thus, the HFD produced the highest epididymal fat deposit, which was reduced most noticeably with the addition of 10% or 20% E over time, regardless of hepatic miR-29a expression.

### 2.4. Serum Biochemical Parameters

In addition to body and organ characteristics, biochemical parameters, including AST and ALT, were also investigated (Figure 4). Short-term (4 weeks) feeding revealed no effect on AST, but differences became apparent beginning at 9 weeks (Figure 4A). Significant increases in AST were observed in the HFD (*p* < 0.001) and HFD plus 10% E (*p* < 0.0001) and 20% E (*p* < 0.05) at 26 weeks relative to the CMD. Increased activity was observed in the HFD + 2% E at 39 weeks relative to the CMD (*p* < 0.01). An inverse relationship between the EtOH concentration and AST level was observed at 39 weeks.

Similarly, an overall trend of increased ALT was observed throughout the time course (Figure 4B). At 26 weeks feeding, the lowest ALT level was found in the CMD group, while the ALT levels in the HFD (*p* < 0.0001), HFD + 10% E (*p* < 0.01), and 20% E (*p* < 0.01) were significantly higher. The HFD ALT remained significantly higher at 30 weeks (*p* < 0.05 for HFD + 2% and 10% E; *p* < 0.0001 for HFD + 20% E). An inverse relationship was observed between the EtOH concentration and ALT level at 39 weeks, with the HFD + 20% E group level measuring particularly low (*p* < 0.0001) relative to the CMD group (Figure 4B). 

The especially low ALT level in the HFD + 20% E group resulted in it possessing the highest AST/ALT ratio at 39 weeks, an indicator of having the worst liver condition relative to the other groups (Figure 4C). Interestingly, 2% E had a continuous trend of increasing AST and ALT, which was opposite to the other groups. The correlation between hepatic miR-29a expression and liver enzymes showed that lower miR-29a expression was associated with a trend toward higher AST and significantly higher ALT (*p* < 0.01), but a lower AST:ALT ratio. Collectively, increased AST and ALT levels were observed in all groups at multiple time points, indicating liver damage with long-term feeding and EtOH concentration specificity. Down-regulation of miR-29a was associated with higher liver enzyme levels.

### 2.5. Histology Analysis

To examine the extent of the liver damage at a microscopic level, Oil Red O, Tri-Chrome, and H&E staining were performed. Representative liver histological images are shown in Figure 5, Figure 6 and Figure 7. After 13 weeks, liver pallor was obvious in the HFD group, less so in the HFD + E groups. No such pallor was apparent in the CMD group (Figure 5A). Lipid deposits were increased in the HFD-containing groups when compared to the CMD group, with a trend toward a higher lipid area in the HFD and HFD + 10% E groups (Figure 5B and Figure 8A). However, no livers showed obvious signs of fibrosis or ballooning at 13 weeks, with only occasional mild inflammation in the HFD and HFD + 10% E groups, and mild perisinusoidal fibrosis in the HFD + 20% E group. After 26 weeks, all livers appeared pale, and lipids were significantly increased in all groups relative to the CMD (Figure 6B and Figure 8A). All groups possessed a degree of fibrosis except the CMD (Figure 9B). After 39 weeks, liver appearances displayed a pallor relative to 13 weeks (Figure 7A), however lipid deposits decreased in all the groups. Importantly, fibrosis became obvious in all the groups, though the HFD, HFD + 2% E, and HFD + 10% E groups possessed relatively higher estimated fibrosis scores (Figure 9B). 

Densitometric analysis of the Oil Red O images confirmed the peak lipid area at 26 weeks (Figure 8A). The highest proportion of lipid droplets in all the groups fell into three size categories, 0–5, 5–100, and 100–5000 μm^2^ (Figure 8B). Using the CMD groups as baselines for each time point, the relative abundances of each droplet type by size were plotted for the four HFD groups (Figure 8C). All HFD-containing groups exhibited decreased small- and medium-size lipid droplets but increased large-size lipid droplets relative to the CMD at 26 weeks, indicating less protective (small size) but more detrimental (large size) lipid droplets in the HFD-containing groups (Figure 8C). While the small- and medium-size lipid droplets saw an increase after 39 weeks of feeding, which is consistent with the better liver histology shown above, the large-size droplets remained relatively low compared to 13 weeks. No change was found in the HFD + 10% E and HFD + 20% E groups from 26 to 39 weeks, but the HFD and HFD + 2% E groups displayed increased large-size lipid droplets in this period (Figure 8C).

Blind assessments of the liver tissue sections by a pathologist revealed that the CMD had the lowest inflammation and fibrosis scores, however the inflammation scores were almost doubled in all groups at 39 weeks feeding when compared to 26 weeks, with the HFD and HFD + 2% E groups possessing a higher degree of inflammation than the other groups (Figure 9A). A negative correlation was found between miR-29a and inflammation (*p* < 0.01) (Figure 9D). Interestingly, all groups except the HFD + 2% E have significantly higher fibrosis scores when compared to the CMD at 39 weeks (Figure 9B) and a negative correlation was found between miR-29a and fibrosis (*p* < 0.01) (Figure 9E). Notably, the HFD and HFD + 2% E groups have lower peak miR-29a induction and higher inflammation and fibrosis scores at 39 weeks than the other groups, indicating a protective role of miR-29a in liver disease development. At 39 weeks, the 20% E group also showed lower microvesicular steatosis (MiS) scores (Figure 9C). However, no significant correlation was found between miR-29a and the MiS scores (Figure 9F). These data are in line with other biochemical measurements and indicate that no concentration of ethanol is particularly protective against HFD-induced liver inflammation and fibrosis. 

### 2.6. Overall Features

We assessed other features, including the appearance of dermatitis, liver neoplasms, and mortality (Appendix A). Representative neoplasm-bearing liver photographs are included in Appendix A. From the liver appearance, several of the specimens possessed multiple neoplasms, suggestive of cirrhosis. The occurrence rate of neoplasia was approximately 7% in the HFD group and 2% in the EtOH-containing groups (Appendix A). EtOH-induced neoplasia has been described in previous research [51], though it is intriguing why a HFD alone produced more instances in this study. We observed premature death only in the HFD and HFD + 20% E groups (2% occurrence rate). Additionally, some mice developed dermatitis and displayed scratching and lesions after long-term adherence to the diets (Appendix A). Curiously, the 2% E group had a four times higher rate of dermatitis than the other groups. In humans, pruritus is common in cholestatic liver disease and perhaps explains the observed phenomenon [52]. Ulcerative dermatitis is also a common occurrence in C57BL/6 mice, which might be exacerbated by these diets [53]. 

## 3. Discussion

Compared to the CMD group, HFD and EtOH consumption increase the risk of the development of liver disease. Given its central role in ethanol and fat metabolism, the liver is uniquely positioned for HFD- and EtOH-induced tissue injury. However, emerging evidence suggests that the extrahepatic effects of ethanol and excessive fat consumption are also critical for providing the additional insults necessary for liver disease morbidity and mortality. To the best of our knowledge, this is the first long-term study to treat animals with a HFD plus different concentrations of EtOH, sampling over four different time points, and demonstrating liver fibrosis. Compared to a HFD alone, a HFD + EtOH feeding does not protect against HFD-induced liver damage, although the addition of EtOH produces concentration-dependent effects after long-term feeding in various measurements. We found that short-term (4 weeks) HFD and EtOH consumption had little effect on BW, liver weight, or AST and ALT measurements. This is an important observation for establishing a minimum feeding length for future research. However, with increased lengths of feeding (26 weeks and 39 weeks), the differences in these measurements were clearly evident. Overall, the addition of 2% E results in a relatively high level of fibrosis after 39 weeks of feeding. The aged group (39 weeks of feeding) showed less hepatic fat when compared to other time points. Overall, hepatic miR-29a expression was negatively associated with the fibrosis score (*p* < 0.01), inflammation score (*p* < 0.01), and liver weight:BW ratio (*p* < 0.05). We conclude that the addition of EtOH results in concentration-dependent liver damage, and that lower levels of hepatic miR-29a induction at early time points are associated with worse liver damage at the endpoint of the study.

Chronic moderate alcohol consumption has been demonstrated to either ameliorate HFD-induced liver fibrosis or aggravate HFD-induced steatohepatitis [31,32]. The conflicting findings are partially due to the different EtOH administration methods applied, such as gavage, consumption of a liquid diet with EtOH, or EtOH in water. The gavage method may induce stress in animals and liquid diets often result in body weight loss. Consumption of EtOH in water is moresimilar to the way in which alcohol is consumed in humans. This position is corroborated by a study using crossed high alcohol-preferring (cHAP) mice, which were selectively bred for high EtOH consumption [54]. In that study, even though voluntary consumption of 10% E produced a higher blood EtOH level than that of animals given the Lieber-DeCarli liquid diet + EtOH (5%), the liver damage of cHAP mice remained lower in the voluntary consumption group [55]. It might be the case that liquid diet feeding has adverse effects on health compared to voluntary consumption with access to dry food.

Additionally, prior animal studies have been performed over much shorter lengths of feeding than the current study, only eliciting liver steatosis, not fibrosis. One significant finding is that liver fibrosis can be induced when feeding is administered for 26 weeks and onward, proceeding to advanced fibrosis with longer feeding. At the 26-week feeding length, most liver fibrosis scores were estimated at the F1 stage with mild fibrosis. By 39 weeks, most mice reached the F2 stage with moderate fibrosis or F3 stage with severe fibrosis (Figure 9B). Our unexpected finding is that the HFD + 2% E group demonstrated an increase in liver fibrosis (*p <* 0.0001), liver inflammation (*p* < 0.05), and higher AST (*p* < 0.01) compared to the CMD group after 39 weeks. 

Additionally, an inverse relationship between the EtOH concentration and liver fibrosis scores was detectable at 39 weeks, indicating that EtOH concentration, length of feeding, and aging all factor in the observed liver phenotype. It seems that 20% E might reduce liver fibrosis in the aged group, although one mouse with liver neoplasia and one premature death were also among this group. Thus, HFD + 20% E feeding may cause decreased body weight, increased lipid oxidation, and energy expenditure [56,57], resulting in a mechanism of enhanced liver damage distinct from that of the HFD + 2% E feeding group. 

The interaction between diet and EtOH may differentially affect inflammation, epigenetic changes, and intestinal microbiota, all of which can impact liver injury [55,58,59,60]. In our experiment, the combinatorial effect of EtOH and a HFD is a far more complicated and nonlinear relationship, with resulting liver injuries dependent upon diet, feeding length, and age. We hypothesize that different levels of EtOH intake may result in divergent cellular events, leading to different routes of liver damage. Similarly, moderate alcohol intake showed a negative association with steatohepatitis and a positive association with hepatocellular carcinoma in NAFLD [29]. Specifically, we have discovered that miR-29a expression responds to a HFD plus EtOH with dosage specificity, and that lower miR-29a expression in 2% E after 13 weeks of feeding is associated with higher liver fibrosis after 39 weeks of feeding. Thus, divergent miR-29a induction indicates distinct liver damage outcomes.

A HFD plus EtOH consumption has been shown to cause liver injury and trigger an inflammatory response [34,35]. With increasing lengths of feeding in this study, the progression of liver damage is evident. However, we noticed an inflection point after 26 weeks, reflected by multiple outcome measures, including biochemistry, histology, and weight. Long-term feeding with a HFD was associated with a decline in the percentage of BW gain, liver weight, serum AST and ALT, and lipid area. We speculate that a HFD and EtOH consumption-induced damage, in combination with aging effects, results in a homeostatic shift, which began earlier in the HFD + 20% E group. Interestingly, long-term (26 weeks) consumption of a HFD with or without EtOH resulted in increased inflammation scores from 26 weeks to 39 weeks of feeding. One explanation for the observed increase in inflammation, in addition to the effect of aging, is that a HFD contains polyunsaturated fatty acids, which are highly susceptible to peroxidation and can adversely impact membrane composition and increase cellular oxidative stress [61]. Ethanol plays important roles in lipid membrane properties, including lipid size and function [62,63,64,65]. The mechanisms by which EtOH benefits HFD-induced liver damage and enzyme alteration remain to be fully elucidated, but may involve the regulation of glucogenesis, lipid metabolism, or microbiota dysbiosis [66,67]. At the 26-week feeding length, we saw a dissociation between inflammation and fibrosis, such that higher EtOH intake groups (10% and 20%) exhibited higher inflammation without displaying as high fibrosis scores. A similar dissociation between inflammation and liver fibrosis was found in a study of rats fed a high-fat, high-cholesterol (HFHC) diet supplemented with Chinese spirits. Chinese spirits and pure ethanol did not improve hepatic inflammation, but did ameliorate HFHC-induced activation of Kupffer cells and hepatic stellate cells [31]. Thus, long-term consumption of different concentrations or types of alcohol may have distinct protective/exacerbating effects on liver disease. However, compared to the CMD group, which has higher body weight gain and less liver damage, both EtOH and a HFD are evidentially detrimental to liver health.

Dynamic alteration of miR-29a was observed with peak expression after 13 weeks, increasing to 7, 5, and 3 times for the 4-week feeding level in the HFD + 10% E, + 20% E, and + 2% E groups, respectively. However, peak miR-29a expression in the HFD group occurred after 26 weeks and was only twice the level of the 4-week feeding. This suggests that the addition of EtOH enhances miR-29a up-regulation. At 39 weeks, significant down-regulation of miR-29a was observed in all the groups, in association with a high fibrosis burden. The role of miR-29a in liver disease is profoundly important, as it is involved in liver steatosis, fibrosis, and severe cholestasis-induced early liver fibrosis. The correlation between hepatic, serum, and circulating extracellular vesicular expressions of miR-29a and liver disease severity has positioned miR-29a as a potential biomarker and drug target [68]. Multiple studies found lower expression of miR-29a in livers from patients with advanced liver fibrosis. One possible mechanism is TGF-β- and NF-κB-dependent down-regulation of miR-29a and commensurate HSC activation. Inhibition of miR-29a results in the up-regulation of α-SMA, *Col1*, and *Fstl1* [69]. Interestingly, increased miR-29a was also found to ameliorate HFD-induced NASH in mice [70]. Another study demonstrated that people with type 2 diabetes mellitus (T2DM) have increased miR-29a, suggesting that miR-29a may function to counter increased fat in the liver. The same study found that increased miR-29a is associated with alcoholism [71]. The mechanism driving EtOH- or fat-induced miR-29a expression is still unknown. In this study, the expression of miR-29a demonstrated an inverse U-shaped relationship with EtOH concentration at 13 weeks. Importantly, the 2% E group had a relatively lower miR-29a level, which was associated with the greatest degree of liver fibrosis in the long run. Over the course of the study, miR-29a demonstrated dynamic alterations, which were associated with liver disease progression.

When comparing a high-fat diet (HFD) with a chow diet, the effect of dietary fat is a variable, and few studies currently include a matched low-fat control diet in comparison to a purified ingredient high-fat diet. Comparing a HFD to a CMD enabled the identification of the long-term feeding effects of these disparate diets on body weight, and provided insight on a heated discussion as to whether carbohydrates or fats are most relevant to promoting body weight gain. Relative to a CMD, HFD feeding resulted in higher average body weights after feeding for about 16 weeks. Interestingly, this is an inflection point for body weight change (Figure 1B, gray box area). Rather than increasing, the HFD group body weight gains flattened, as did those of the HFD plus 2% and 10% E groups. After 30 weeks, the body weights consistently decreased in all HFD plus EtOH groups and the HFD plus water group, reaching a significant difference when compared to the CMD from 33 weeks onward. Similarly, the percentage of body weight gain was higher in the HFD than the CMD when treated for less than 26 weeks, but lower after this point (Appendix A). Even though the CMD group gained the most BW, the liver parameters revealed lower inflammation, fibrosis, and MiS scores at 26 weeks treatment (Figure 9A). The data indicate that animals from the HFD or HFD plus EtOH groups have worse overall liver health than the CMD in the long term.

In summary, our feeding design more accurately mimics human alcohol consumption, and the time course of the study provides valuable information about the combination of a HFD and EtOH consumption development and with aging. Our study indicates that the combined effects result in significant liver damage only with longer feeding duration, producing liver fibrosis and worse general health. The addition of 2% E results in higher liver fibrosis, while the addition of 20% E results in the worst overall health outcomes. Aging in combination with various feeding paradigms results in a homeostatic shift, that will need to be expanded upon in future studies, at the molecular level in multiple organs. Feeding length and physiological alterations are not linear relationships, highlighting the importance of the age/feeding interplay. Given the multiple variables involved in liver disease, future clinical interventions may need to be tailored to the individual patient’s diet, age, and level of alcohol intake.

## 4. Material and Methods

### 4.1. Animal and Experimental Design

The experimental protocol used in this study was reviewed and approved by the Indiana University Institutional Animal Care and Use Committee and was carried out in accordance with the NIH Guide for the Care and Use of Laboratory Animals. The animals were maintained in facilities fully accredited by the Association for the Assessment and Accreditation of Laboratory Animals. Male C57BL/6 mice (age 8 weeks) were divided into 5 feeding groups (N = 10–12 per group): calorie-matched diet (CMD) plus water, high-fat diet (HFD) plus water, HFD plus obligatory 2% (*v*/*v*) EtOH (2% E) administered in drinking water, HFD plus 10% E, and HFD plus 20% E. The lengths of feeding were 4 weeks, 13 weeks, 26 weeks, and 39 weeks, which corresponds to 12, 21, 34, and 47-week-old mice at the four tissue collection time points (Figure 1A). The corresponding human age range was calculated based on the previous research and online calculating tools (http://www.age-converter.com/mouse-age-calculator.html, accessed on on 20 August 2023), which was approximately teens to 50s (Figure 1A). The mice were housed at 21 °C with a 12 h light–dark cycle and ad libitum access to food and water/ethanol.

The weight of the mice and volumes of liquid consumed were collected twice per week. Serum samples were collected at one time point during and at the end of feeding through facial vein or trunk blood, respectively (Figure 1A, red arrows). Tissues, including liver, epididymal fat, and blood tissues were collected at the end of feeding. Samples were fixed in 10% neutral buffered formalin or frozen fresh at −80 °C. 

### 4.2. Diet and Overall Health

A CMD (TD.110196) and HFD (TD. 06303, 22% HVO) were purchased from Teklad (Envigo, Indianapolis, IN, USA). A CMD contains approximately 15.1% kcal from fat, 17.6% from protein, and 67.3% from carbohydrate, giving this diet 6.2% fat and 62% carbohydrate by weight. A HFD contains approximately 45.2% kcal from trans-fat, 17.7% from protein, and 37.1% from carbohydrate, giving this diet approximately 23.2% fat and 42.9% carbohydrate by weight. The detailed formulae are included in the Appendix A. Overall, no obvious health issues were observed, although some animals exhibited dermatitis after long-term feeding. The skin conditions were treated according to the IACUC guidelines.

### 4.3. Sample Collection and Histological Analysis

Frozen liver tissue histological specimens were blocked, sectioned, and stained with Oil Red O (ORO) and hematoxylin-eosin (H&E). Formalin-fixed liver tissues were paraffin-embedded, sectioned, and stained with Tri-Chrome and H&E. Slides were reviewed and scored in a blinded fashion by a single pathologist. All livers were assessed for the degree of inflammation, fibrosis, and microvesicular steatosis (MiS) using standard criteria [72]. In addition, the lipid area was quantified in randomly selected fields with an equal number of fields for each group. Original images were processed through thresholding, binary conversion, and despeckling, and *ImageJ* (version 1.54f) was used to quantify the lipid droplet size and number, as well as the total lipid area [73]. 

### 4.4. Serum Analysis

Immediately after collection, the blood samples were kept on ice and allowed to coagulate prior to centrifugation for 15 min at 1000× *g*. Serum was collected and stored at −80 °C. Activities of aspartate aminotransferase (AST) and alanine aminotransferase (ALT) were measured with a VERSAmax tunable microplate reader (Molecular Devices, San Jose, CA, USA) using colorimetric assay kits (AST, catalog number K753; ALT, catalog number K752, from BioVision, Inc., Milpitas, CA, USA). 

### 4.5. RNA Isolation

Total RNA was extracted from snap frozen tissue samples using NucleoSpin miRNA mini kits (catalog number 740971 from Macherey-Nagel, Inc., Allentown, PA, USA), following the manufacturer’s instructions. The concentration and purity of the extracted RNAs were measured using a Nanodrop 2000 spectrophotometer (Thermo Fisher Scientific, Carlsbad, CA, USA). RNA was reverse transcribed to cDNA using the SuperScript III First-Strand Synthesis System (18080051, Thermo Fisher Scientific, Carlsbad, CA, USA). To measure mature miR-29a expression, reactions were set up utilizing TaqMan Fast Advanced Master Mix (4444557, Applied Biosystems, Foster City, CA, USA) with a TaqMan probe and primers for mature miR-29a (478587_mir, Applied Biosystems, Foster City, CA, USA), or U6 snRNA (001973, Applied Biosystems, Foster City, CA, USA). Moreover, miRNA expression was normalized to U6. Samples were run in duplicate with a 10 µL total volume on an ABI 7500 real-time PCR machine (Applied Biosystems, Foster City, CA, USA). Relative expressions were analyzed using the ^ΔΔ^CT method. 

## Figures and Tables

**Figure 1 ijms-24-14564-f001:**
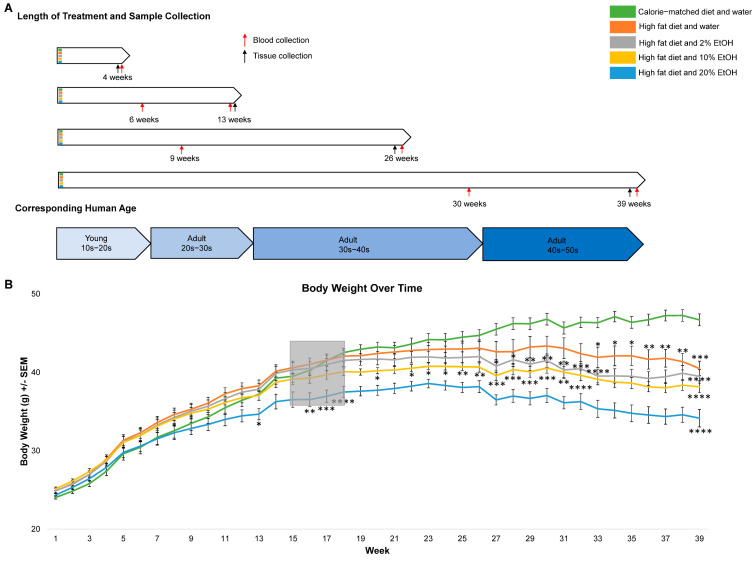
Experimental design and body weight. (**A**) Overview of feeding groups, lengths of feeding, and sample collection points. Blood collections for serum analyses indicated with red arrows. Tissue collections for histology indicated with black arrows. Approximate human ages are presented at the bottom. (**B**) Average weekly BW of all mice in the study plotted over 1–39 weeks. Graph represents mean values +/− SEM for each week. After each tissue harvesting time point, available animal numbers decrease. From week 1–4, N = 42–43/group. From week 5–13, N = 30–31/group. From week 14–26, N = 20–21/group. From week 27–39, N = 10–11/group. Comparisons were made using a two-way ANOVA, followed by Dunnett’s multiple comparisons test. Comparisons made against the CMD group for a given time point. Statistical significance was labeled as follows: * as *p* < 0.05, ** as *p* < 0.01, *** as *p* < 0.001, and **** as *p* < 0.0001.

**Figure 2 ijms-24-14564-f002:**
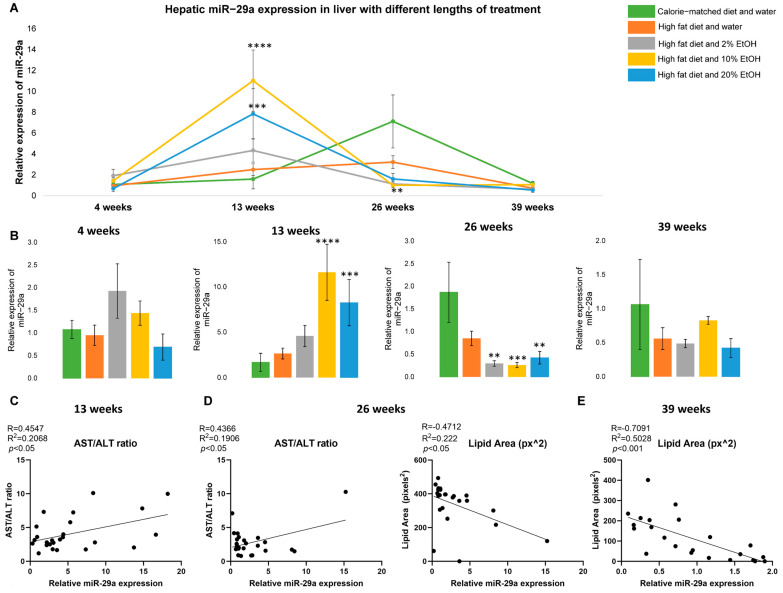
Hepatic miR-29a expression in liver. (**A**) Relative hepatic miR-29a expression in mice fed with a CMD, HFD, HFD + 2% E, HFD + 10% E, and HFD + 20% E for 4, 13, 26, and 39 weeks. Comparisons were made using a two-way ANOVA, followed by Dunnett’s multiple comparisons test. Comparisons made against the CMD group for a given time point. (**B**) Hepatic miR-29a expression relative to a CMD at 4, 13, 26, and 39 weeks. Significant correlation between miR-29a expression and (**C**) AST/ALT ratio at 13 weeks, (**D**) AST/ALT ratio and lipid area at 26 weeks, and (**E**) lipid area at 39 weeks of feeding. Correlations were performed, generating Pearson r values. This was followed by a simple linear regression. Statistical significance was labeled as follows: ** as *p* < 0.01, *** as *p* < 0.001, and **** as *p* < 0.0001. Each black dot represents data from an individual sample.

**Figure 3 ijms-24-14564-f003:**
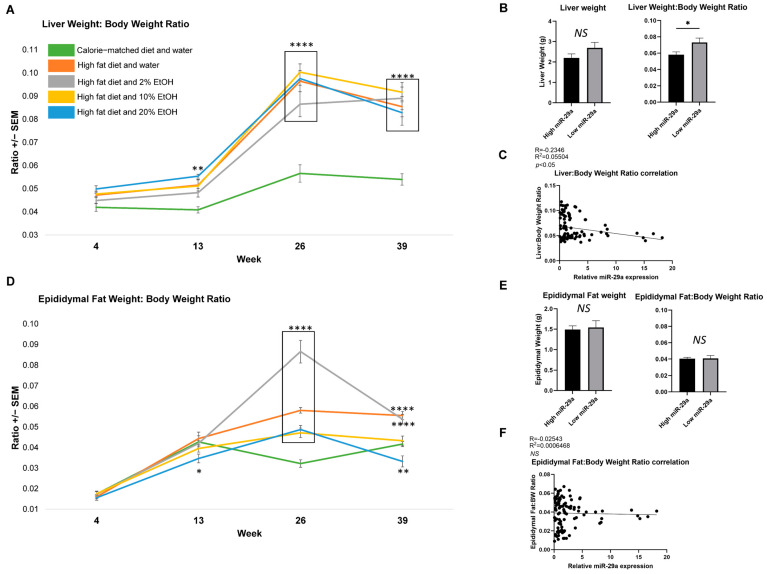
Average liver weight:BW ratios and epididymal fat weight:BW ratios over time. (**A**) Average liver weight:BW ratios plotted for 4, 13, 26, and 39-week groups. Graphs represent mean values +/− SEM, with N = 9−12/group for each time point. Comparisons were made using a two-way ANOVA, followed by Dunnett’s multiple comparisons test. Comparisons made against the CMD group for a given time point. (**B**) Comparisons between the 25 samples with the highest and 25 samples with the lowest miR-29a expression (i.e., top and bottom 25% of the samples) revealed differences in the liver weight:BW ratio. Comparisons were made using Student’s *t*-test, with the statistical significance labeled as follows: * as *p* < 0.05. (**C**) A negative correlation was found between miR-29a expression and the liver weight:BW ratio. (**D**) Average epididymal fat weight:BW ratios plotted for 4, 13, 26, and 39-week groups. The average epididymal fat weight:BW ratios increase over time and demonstrate an inverse correlation with ethanol concentration after 39 weeks of feeding. Graphs represent mean values +/− SEM, with N = 9–12/group for each time point. Comparisons were made using a two-way ANOVA, followed by Dunnett’s multiple comparisons test. Comparisons made against the CMD group for a given time point. (**E**) Comparisons between the 25 samples with the highest and 25 samples with the lowest miR-29a expression (i.e., top and bottom 25% of the samples) revealed no differences in the average epididymal fat weight and the epididymal fat weight:BW ratio. (**F**) No correlation was found between miR-29a expression and the epididymal fat weight:BW ratio. Correlations were performed, generating Pearson r values. This was followed by a simple linear regression. Statistical significance labeled as follows: * as *p* < 0.05, ** as *p* < 0.01, and **** as *p* < 0.0001. NS: not significant. Each black dot represents data from an individual sample.

**Figure 4 ijms-24-14564-f004:**
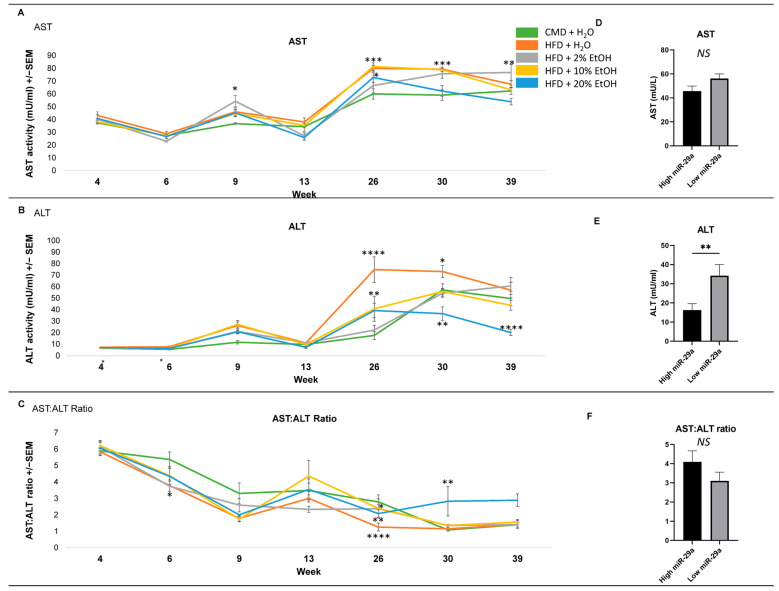
Average AST, ALT, and AST:ALT ratios over time and their association with miR-29a expression. (**A**) Average AST and (**B**) ALT activity (mU/mL) and (**C**) AST:ALT ratios plotted for each collection time. With the exception of the 4-week feeding length, blood samples were collected at two time points for all mice. Line graphs reflect average measurements over time. Graphs represent mean values +/− SEM, with N = 9–12/group for each time point. Comparisons were made using a two-way ANOVA, followed by Dunnett’s multiple comparisons test. Comparisons made against the CMD group for a given time point. (**D**–**F**) Comparisons between the 25 samples with the highest and 25 samples with the lowest miR-29a expression (i.e., top and bottom 25% of the samples) revealed a negative correlation between miR-29a expression and ALT activity. Comparisons were made using Student’s *t*-test. Statistical significance labeled as follows: * as *p* < 0.05, ** as *p* < 0.01, *** as *p* < 0.001, and **** as *p* < 0.0001. NS: not significant.

**Figure 5 ijms-24-14564-f005:**
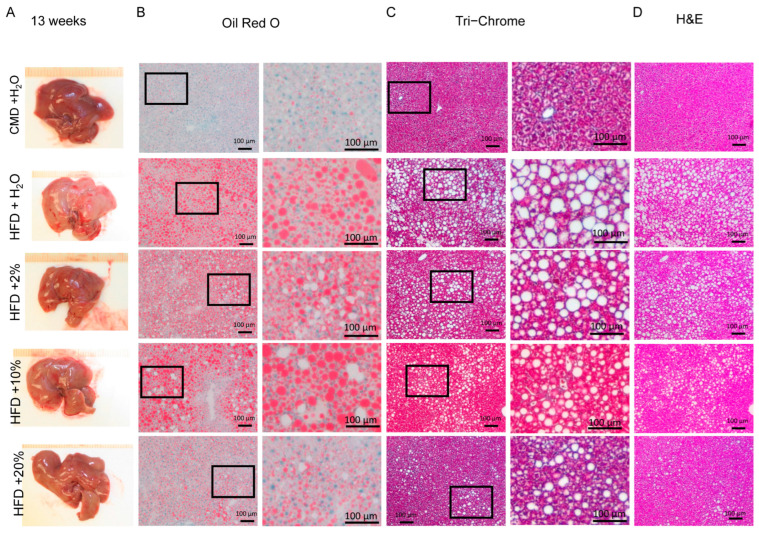
Liver histology at 13 weeks. (**A**) Representative gross morphology of livers from each feeding group. (**B**) Oil Red O staining, with enlarged area indicated by box, from representative liver sections. (**C**) Tri-Chrome staining, with enlarged area indicated by box, from representative liver sections. (**D**) Hematoxylin and eosin staining from representative liver sections.

**Figure 6 ijms-24-14564-f006:**
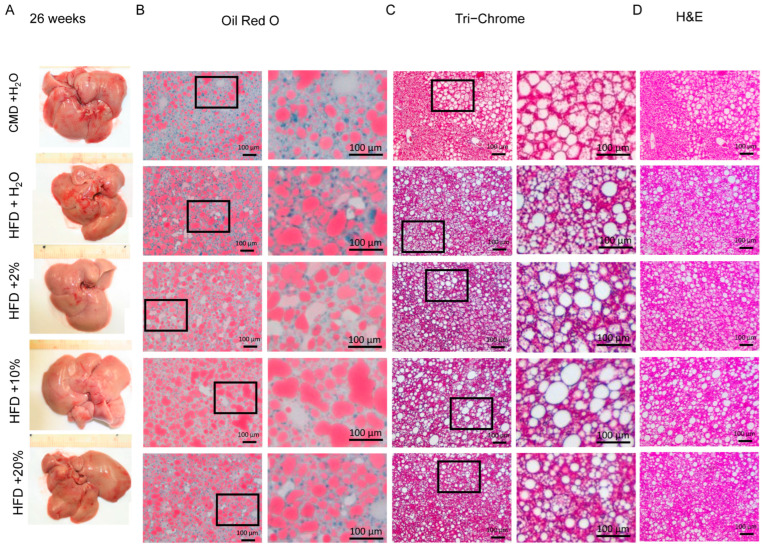
Liver histology at 26 weeks. (**A**) Representative gross morphology of livers from each feeding group. (**B**) Oil Red O staining, with enlarged area indicated by box, from representative liver sections. (**C**) Tri-Chrome staining, with enlarged area indicated by box, from representative liver sections. (**D**) Hematoxylin and eosin staining from representative liver sections.

**Figure 7 ijms-24-14564-f007:**
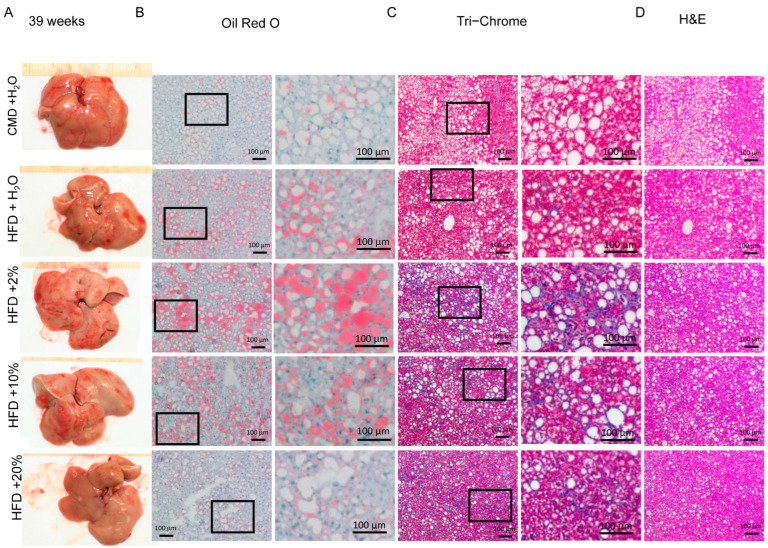
Liver histology at 39 weeks. (**A**) Representative gross morphology of livers from each feeding group. (**B**) Oil Red O staining, with enlarged area indicated by box, from representative liver sections. (**C**) Tri-Chrome staining, with enlarged area indicated by box, from representative liver sections. (**D**) Hematoxylin and eosin staining from representative liver sections.

**Figure 8 ijms-24-14564-f008:**
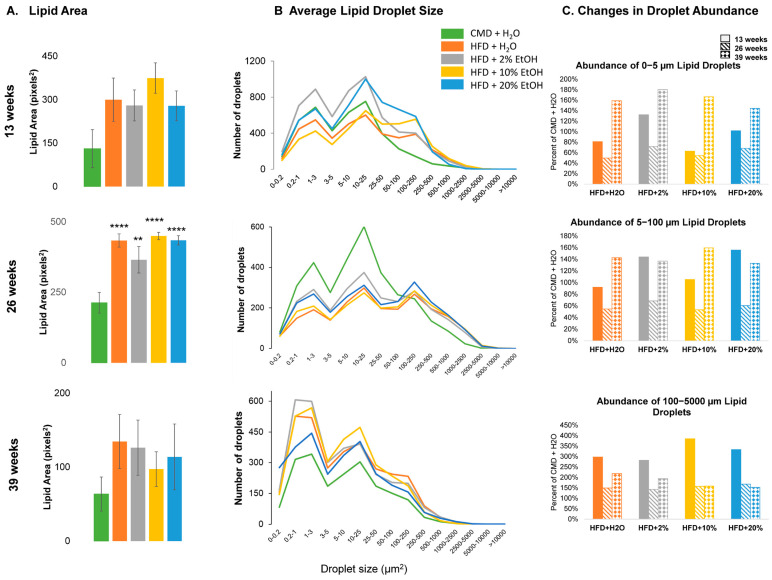
Total lipid area and droplet size analysis. (**A**) Average lipid area as determined by densitometric analysis of Oil Red O photomicrographs. Graphs represent mean values +/− SEM, with N = 7/group. Comparisons were made using a one-way ANOVA, followed by Dunnett’s multiple comparisons test. Comparisons made against the CMD group for a given time point. (**B**) Abundance of lipid droplets of particular sizes. Graphs represent the average number of droplets of a given size counted for each feeding group. (**C**) Change in the relative proportions of different sized lipid droplets over time. Graphs represent the percent abundance of particular sized lipid droplets relative to the CMD group at the same time point. Statistical significance labeled as follows: ** as *p* < 0.01, and **** as *p* < 0.0001.

**Figure 9 ijms-24-14564-f009:**
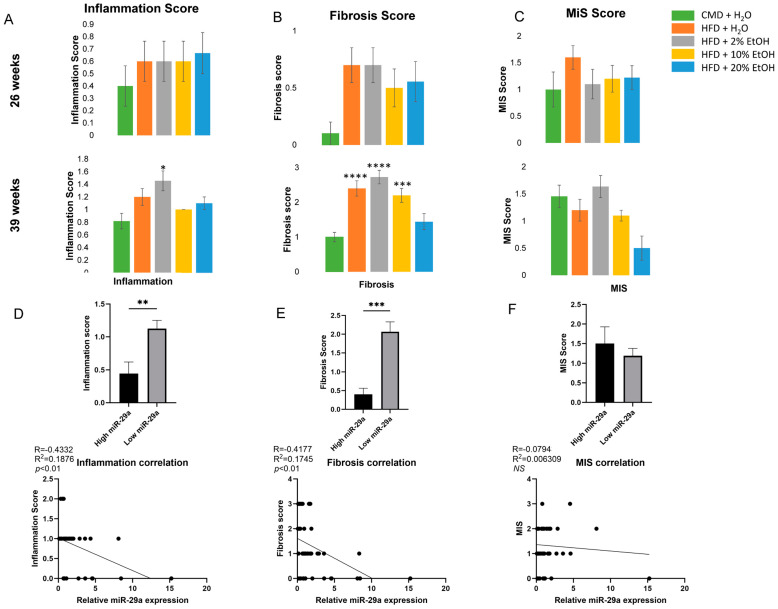
Histological assessments and correlation with miR-29a at 26 and 39 weeks. (**A**) Average inflammation scores for the feeding groups at 26 and 39 weeks, as determined by a professional histopathologist. Graphs represent mean values +/− SEM, with N = 6–11/group. (**B**) Average fibrosis scores for the feeding groups at 26 and 39 weeks, as determined by a professional histopathologist. Graphs represent mean values +/− SEM, with N = 5–11/group. (**C**) Average microvesicular steatosis (MiS) scores for the feeding groups at 26 and 39 weeks, as determined by a professional histopathologist. Graphs represent mean values +/− SEM, with N = 4–10/group. Comparisons were made using a one-way ANOVA, followed by Dunnett’s multiple comparisons test. Comparisons made against the CMD group for a given time point. The 25 samples with the highest and 25 samples with the lowest miR-29a expression (i.e., top and bottom 25% of the samples) demonstrated a significant difference in (**D**) inflammation and (**E**) fibrosis, but not (**F**) the MiS score. Graphs represent average values +/− SEM for the given traits. Comparisons were made using Student’s *t*-test. Moreover, miR-29a expression is negatively correlated with (**D**) inflammation (N = 40) and (**E**) fibrosis (N = 41) scores, but not with (**F**) the MiS scores. Correlations were performed, generating Pearson r values. This was followed by a simple linear regression. Statistical significance for all graphs was labeled as follows: * as *p* < 0.05, ** as *p* < 0.01, *** as *p* < 0.001, and **** as *p* < 0.0001. Each dot represents data from an individual sample.

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
