# Peer review of "Dynamic Alterations to Hepatic MicroRNA-29a in Response to Long-Term High-Fat Diet and EtOH Feeding"

_ijms, 2023, doi:10.3390/ijms241914564_

Round 1

Reviewer 1 Report

This study of Williams et.  showed interesting results on hepatic miRNA-29a using HFD + EtOH fed mice. However, major and minor concerns are needed to be addressed:

1. The main title and the short title do not match. These titles are somewhat different and the authors may need to decide which one is more suitable according to the results and conclusion.
2. Did the authors conduct this experiment with a normal control group? My main concern in this study is the absence of the normal control. All results have no basis or comparisons to the normal mice without HFD and EtOH-fed. Hence, the markers and parameters tested levels may not be sure whether it has changed from and back to the NC. Providing normal control will also compare the effect of aging to normal mice which the authors claimed have significant results in the HFD and EtOH-fed mice.

3. What are the basis of the different EtOH concentrations in the experiment? Similarly, how did the authors come up with the time period of the experiment? In addition, the authors provided the equivalence of the time period according to human ages, what was the basis of this?/ please provide citations.
4. The asterisk or significance in the figures especially in line graphs with multiple comparisons are not well defined to which group or time it is compared with. 
5. In Figure 2: 2B did not show significant difference in each group? It just seems that the bar graph are statistically significant. 2C, 2D and 2E- It is somewhat confusing why different marker are tested at different time points. it may mean that these markers are the only results that showed significance but would it be possible to show these markers at all time points to clearly see how these markers are correlated .

6.What does arrow in figures 5 and 6 mean? Similar figure in figure 7 did not indicate an arrow.
7.  Interesting results on 2% alcohol effect is still unclear for me and may need more explanation in discussion part.
8. Mortality of mice have been shortly discussed but I still wonder at long term feeding of 39 weeks, how many mice died? As explained by the authors the traditional way of alcohol induction esp to yield alcohol - induced fibrosis in mice usually result to not obtaining fibrosis and/or high mortality rate of mice. The n in each time point of experiments may be needed to explain in figure legends 
9. I am a little concerned about the use of ethanol or alcohol treatment in the whole manuscript- it may mean alcohol as a treatment group- consider using other words to clearly indicate the desired content - example EtOH-fed
10. Please be careful and mindful of spelling, typographical mistakes, and some grammar mistakes - line 200, line 220-221, line 236, line 305, line 308, line 420, line 456, line 458, line 475 and others I may have not seen too. These are minor mistakes and may be proofread later but still needs to be carefully checked. 

See comment 10.

Author Response

We appreciate all the comments from reviewers, and the quality of paper was improved. Due to the addition of the CMD group, new analyses have been carried out and all the figures have been revised. More details are included in the following point-to-point answering letter.

Reviewer 1.

This study of Williams et. al showed interesting results on hepatic miRNA-29a using HFD + EtOH fed mice. However, major and minor concerns are needed to be addressed:

  1. The main title and the short title do not match. These titles are somewhat different and the authors may need to decide which one is more suitable according to the results and conclusion.

Answer: We appreciate your comment, and the title was changed to “Dynamic Alterations of Hepatic MicroRNA-29a in Response to Long-term High Fat Diet and Alcohol Feeding”

  1. Did the authors conduct this experiment with a normal control group? My main concern in this study is the absence of the normal control. All results have no basis or comparisons to the normal mice without HFD and EtOH-fed. Hence, the markers and parameters tested levels may not be sure whether it has changed from and back to the NC. Providing normal control will also compare the effect of aging to normal mice which the authors claimed have significant results in the HFD and EtOH-fed mice.

Answer: When the experiment was designed, we included a calorie-matching diet (CMD) group and carried out the experiment at the same time with other groups. We initially planned to publish a separate paper on comparing CMD vs HFD only. Published papers mentioned that few publications included a matched low-fat control diet in comparison to high-fat diet, and some parameters, such as insulin sensitivity and body weight, were affected by the choice of control diet (1, 2). We agree with the reviewer and have included the CMD group in this paper.

  1. What are the basis of the different EtOH concentrations in the experiment? Similarly, how did the authors come up with the time period of the experiment? In addition, the authors provided the equivalence of the time period according to human ages, what was the basis of this?/ please provide citations.

Answer: Based on previous publications and our own 20 years experience in alcohol research, C57BL/6J mice display concentration-dependent responses to ethanol and drink more in both voluntary and schedule-controlled conditions (3, 4). C57BL/6J mice prefer to drink alcohol, with concentrations ranging from 0-30% (v/v). To model low, medium, and high alcohol intake, we elected to use 2%, 10%, and 20% (v/v) ethanol concentrations. More detail will be included in the revised introduction.

We planned to test different lengths of treatment that range from young/adolescent period to mature mice. The lengths of treatment were based on previous studies. Our design is from short term (28 days/4 weeks) to long term (90 days, 180 days, and 270 days which corresponds to 13, 26, and 39 weeks). This design allowed us to have mid-point blood collections while managing sample and data collection for all groups. A recent study found 30 weeks HFD induced metabolic disorders and behavior change that is consistent with our finding (5). 

Living in the ideal condition, the life span of a mouse is approximately 2 years, and the corresponding mouse and human ages have been studied. One day in mice is roughly equivalent to 40 human days, while one human year equates to about 9 mouse days. The reference was added in the revised paper. 

  1. The asterisk or significance in the figures especially in line graphs with multiple comparisons are not well defined to which group or time it is compared with. 

Answer: We have updated the figure legends to reduce this source of confusion.

  1. In Figure 2: 2B did not show significant difference in each group? It just seems that the bar graph are statistically significant. 2C, 2D and 2E- It is somewhat confusing why different marker are tested at different time points. it may mean that these markers are the only results that showed significance but would it be possible to show these markers at all time points to clearly see how these markers are correlated
    .

Answer: We have added the marker on the statistically significant groups. Since figures have been revised and 2C, 2D and 2E in the previous version showed in 2B of new version.  

6.What does arrow in figures 5 and 6 mean? Similar figure in figure 7 did not indicate an arrow.

Answer: The arrows are indicating damage caused by steatosis. They were initially observed only in EtOH containing group at 13 weeks (Figure 5) and were also found in HFD and CMD group. However, these whitened areas are different from those lipid droplets with partial area stained with Oil Red O in HFD, HFD+2% E and 10% E groups. These damaged areas seem different from damaged cells of HFD+20% E group at 39 weeks. We do not fully understand this phenomenon and future characterizations may clarify this initial observation. We elected to not discuss to avoid distraction at this submission. 

  1.  Interesting results on 2% alcohol effect is still unclear for me and may need more explanation in discussion part.

Answer: From the data we collected, at 39 weeks, HFD + 2%E group has a trend of increased AST, and inflammation, which are the main measurements in agreement with the higher liver fibrosis observed. We also studied the lipid size differences and found that HFD+2% E has increased percentage of large size lipid (100-5000 μM2) after 39 weeks treatment when compared to 26 weeks. Our explanation is that large size lipids have detrimental effects on liver due to their inflammatory nature. This is consistent with our observation that HFD+2% E has higher inflammation. We have tried to understand this phenomenon and find limited data on the combinatorial effect of HFD with different concentrations of alcohol or with long-term treatment.

  1. Mortality of mice have been shortly discussed but I still wonder at long term feeding of 39 weeks, how many mice died? As explained by the authors the traditional way of alcohol induction esp to yield alcohol - induced fibrosis in mice usually result to not obtaining fibrosis and/or high mortality rate of mice. The n in each time point of experiments may be needed to explain in figure legends 

Answer:

We have added the information in S- figure 3 on the number of mice with mortality and tumors.

  1. I am a little concerned about the use of ethanol or alcohol treatment in the whole manuscript- it may mean alcohol as a treatment group- consider using other words to clearly indicate the desired content - example EtOH-fed

Answer: Fully agree with reviewer’s insight. We have revised the manuscript and used EtOH-fed, instead of treatment.

  1. Please be careful and mindful of spelling, typographical mistakes, and some grammar mistakes - line 200, line 220-221, line 236, line 305, line 308, line 420, line 456, line 458, line 475 and others I may have not seen too. These are minor mistakes and may be proofread later but still needs to be carefully checked. 

Answer: Really appreciate reviewers’ comments, and language has been improved in this revision.

References

  1. Xiang DD, Liu JT, Zhong ZB, Xiong Y, Kong HY, Yu HJ, et al. MicroRNA-29a-3p Prevents Drug-Induced Acute Liver Failure through Inflammation-Related Pyroptosis Inhibition. Curr Med Sci. 2023;43(3):456-68.
  2. Lin HY, Yang YL, Wang PW, Wang FS, Huang YH. The Emerging Role of MicroRNAs in NAFLD: Highlight of MicroRNA-29a in Modulating Oxidative Stress, Inflammation, and Beyond. Cells. 2020;9(4).
  3. Xu XY, Du Y, Liu X, Ren Y, Dong Y, Xu HY, et al. Targeting Follistatin like 1 ameliorates liver fibrosis induced by carbon tetrachloride through TGF-beta1-miR29a in mice. Cell Commun Signal. 2020;18(1):151.
  4. Dalgaard LT, Sorensen AE, Hardikar AA, Joglekar MV. The microRNA-29 family: role in metabolism and metabolic disease. Am J Physiol Cell Physiol. 2022;323(2):C367-C77.
  5. Jing F, Geng Y, Xu XY, Xu HY, Shi JS, Xu ZH. MicroRNA29a Reverts the Activated Hepatic Stellate Cells in the Regression of Hepatic Fibrosis through Regulation of ATPase H(+) Transporting V1 Subunit C1. Int J Mol Sci. 2019;20(4).
  6. Jampoka K, Muangpaisarn P, Khongnomnan K, Treeprasertsuk S, Tangkijvanich P, Payungporn S. Serum miR-29a and miR-122 as Potential Biomarkers for Non-Alcoholic Fatty Liver Disease (NAFLD). Microrna. 2018;7(3):215-22.
  7. Yang YL, Tsai MC, Chang YH, Wang CC, Chu PY, Lin HY, et al. MIR29A Impedes Metastatic Behaviors in Hepatocellular Carcinoma via Targeting LOX, LOXL2, and VEGFA. Int J Mol Sci. 2021;22(11).
  8. Yang YL, Kuo HC, Wang FS, Huang YH. MicroRNA-29a Disrupts DNMT3b to Ameliorate Diet-Induced Non-Alcoholic Steatohepatitis in Mice. Int J Mol Sci. 2019;20(6).
  9. Rajabi S, Najafipour H, Sheikholeslami M, Jafarinejad-Farsangi S, Beik A, Askaripour M, et al. Perillyl alcohol and quercetin modulate the expression of non-coding RNAs MIAT, H19, miR-29a, and miR-33a in pulmonary artery hypertension in rats. Noncoding RNA Res. 2022;7(1):27-33.

10.       Eguchi A, Lazaro RG, Wang J, Kim J, Povero D, Willliams B, et al. Extracellular vesicles released by hepatocytes from gastric infusion model of alcoholic liver disease contain a MicroRNA barcode that can be detected in blood. Hepatology. 2017;65(2):475-90.

Reviewer 2 Report

The authors of this manuscript have performed an interesting investigation on the impact of a high-fat diet, alcohol consumption, and hepatic miR-29a in mouse models. The study is intriguing, although it could benefit from further comprehensiveness. The findings indicate that a combination of a High Fat Diet and escalating levels of alcohol consumption negatively affected the liver. Additionally, the up-regulation of miR-29a showed a correlation with mitigated liver injury.

 It is worth noting that the study primarily focuses on physiological parameters. Hence, it would be advantageous to include the analysis of all Serum biochemical parameters, such as the LFT test, Total cholesterol, TG, and others.

 If a more comprehensive understanding of the effects of miR-29a is desired, further analysis could be considered. It is well-established that miR-29a serves as an anti-fibrotic and anti-inflammatory molecule in various organs, including the liver. Therefore, conducting further analysis would allow the authors to represent a comprehensive view of miR-29a's impact.

The English of the manuscript is appropriate.

Author Response

We appreciate all the comments from reviewers, and the quality of paper was improved. Due to the addition of the CMD group, new analyses have been carried out and all the figures have been revised. More details are included in the following point-to-point answering letter.

Reviewer 2.

The authors of this manuscript have performed an interesting investigation on the impact of a high-fat diet, alcohol consumption, and hepatic miR-29a in mouse models. The study is intriguing, although it could benefit from further comprehensiveness. The findings indicate that a combination of a High Fat Diet and escalating levels of alcohol consumption negatively affected the liver. Additionally, the up-regulation of miR-29a showed a correlation with mitigated liver injury.

It is worth noting that the study primarily focuses on physiological parameters. Hence, it would be advantageous to include the analysis of all Serum biochemical parameters, such as the LFT test, Total cholesterol, TG, and others.

Answer:  Serum AST and ALT were included as part of serum biochemical parameters, and matched what was observed in liver histology. However, other parameters cannot be tested at this timepoint due to limitations in available material.

If a more comprehensive understanding of the effects of miR-29a is desired, further analysis could be considered. It is well-established that miR-29a serves as an anti-fibrotic and anti-inflammatory molecule in various organs, including the liver. Therefore, conducting further analysis would allow the authors to represent a comprehensive view of miR-29a's impact.

Answer: We fully agree with reviewer, and plan to conduct more comprehensive studies in terms of gene expression to study the combinatorial effect of fatty acids and EtOH on miR-29a using various liver cell types. Our current study is only the first step toward understanding the physiological alterations accrued during long-term treatment.  

References

  1. Xiang DD, Liu JT, Zhong ZB, Xiong Y, Kong HY, Yu HJ, et al. MicroRNA-29a-3p Prevents Drug-Induced Acute Liver Failure through Inflammation-Related Pyroptosis Inhibition. Curr Med Sci. 2023;43(3):456-68.
  2. Lin HY, Yang YL, Wang PW, Wang FS, Huang YH. The Emerging Role of MicroRNAs in NAFLD: Highlight of MicroRNA-29a in Modulating Oxidative Stress, Inflammation, and Beyond. Cells. 2020;9(4).
  3. Xu XY, Du Y, Liu X, Ren Y, Dong Y, Xu HY, et al. Targeting Follistatin like 1 ameliorates liver fibrosis induced by carbon tetrachloride through TGF-beta1-miR29a in mice. Cell Commun Signal. 2020;18(1):151.
  4. Dalgaard LT, Sorensen AE, Hardikar AA, Joglekar MV. The microRNA-29 family: role in metabolism and metabolic disease. Am J Physiol Cell Physiol. 2022;323(2):C367-C77.
  5. Jing F, Geng Y, Xu XY, Xu HY, Shi JS, Xu ZH. MicroRNA29a Reverts the Activated Hepatic Stellate Cells in the Regression of Hepatic Fibrosis through Regulation of ATPase H(+) Transporting V1 Subunit C1. Int J Mol Sci. 2019;20(4).
  6. Jampoka K, Muangpaisarn P, Khongnomnan K, Treeprasertsuk S, Tangkijvanich P, Payungporn S. Serum miR-29a and miR-122 as Potential Biomarkers for Non-Alcoholic Fatty Liver Disease (NAFLD). Microrna. 2018;7(3):215-22.
  7. Yang YL, Tsai MC, Chang YH, Wang CC, Chu PY, Lin HY, et al. MIR29A Impedes Metastatic Behaviors in Hepatocellular Carcinoma via Targeting LOX, LOXL2, and VEGFA. Int J Mol Sci. 2021;22(11).
  8. Yang YL, Kuo HC, Wang FS, Huang YH. MicroRNA-29a Disrupts DNMT3b to Ameliorate Diet-Induced Non-Alcoholic Steatohepatitis in Mice. Int J Mol Sci. 2019;20(6).
  9. Rajabi S, Najafipour H, Sheikholeslami M, Jafarinejad-Farsangi S, Beik A, Askaripour M, et al. Perillyl alcohol and quercetin modulate the expression of non-coding RNAs MIAT, H19, miR-29a, and miR-33a in pulmonary artery hypertension in rats. Noncoding RNA Res. 2022;7(1):27-33.

10.       Eguchi A, Lazaro RG, Wang J, Kim J, Povero D, Willliams B, et al. Extracellular vesicles released by hepatocytes from gastric infusion model of alcoholic liver disease contain a MicroRNA barcode that can be detected in blood. Hepatology. 2017;65(2):475-90.

Reviewer 3 Report

In this study, the authors examined liver injury in animals treated with high fat diets (HFDs) and ethanol (EtOH) and determined levels of miR-29a over the course of the treatment. Although there are some interesting results in the study, there are many concerns for the authors to address to improve the quality of the manuscript.

1.     The title of the study is totally misleading. The authors do not provide any evidence that miR-29a is protective against liver injury. The current data only found some alterations of miR-29a levels with treatment conditions. The authors must perform genetic manipulation of miR-29a in their animal models to illustrate the protective roles of miR-29a.

2.     Why do the authors specifically focus on miR-29a? Are there other microRNAs involved in the liver injury process?

3.     The authors should include experimental groups of animals fed with normal diets and EtOH alone for comparison.

4.     Are the amounts of food consumption different amongst groups of animals in this study? The authors should provide these data and discuss if they could have impact on the results of the study.

5.     Why does miR-29a change in its expression over the course of the treatment? Which tissues/cell types are responsible for the process? The authors should include some in vitro experiments to support their proposed models.

6.     Statistical analyses are missing in many figures (Figure 2B, 8A, 8C, and 9A)

Author Response

We appreciate all the comments from reviewers, and the quality of paper was improved. Due to the addition of the CMD group, new analyses have been carried out and all the figures have been revised. More details are included in the following point-to-point answering letter.

Reviewer 3

In this study, the authors examined liver injury in animals treated with high fat diets (HFDs) and ethanol (EtOH) and determined levels of miR-29a over the course of the treatment. Although there are some interesting results in the study, there are many concerns for the authors to address to improve the quality of the manuscript.

  1. The title of the study is totally misleading. The authors do not provide any evidence that miR-29a is protective against liver injury. The current data only found some alterations of miR-29a levels with treatment conditions. The authors must perform genetic manipulation of miR-29a in their animal models to illustrate the protective roles of miR-29a.

Answer: The title has been revised to reflect the experimental design and main finding. Please also see the answer to the reviewer 1, question 1.

Our intention was primarily to understand physiological alterations in the liver of mice fed with HFD+EtOH. Our second aim was to identify correlations between miR-29a expression and liver damage. However, to fully understand the role of miR-29a in HFD+EtOH, more molecular work must be performed in the future.

  1. Why do the authors specifically focus on miR-29a? Are there other microRNAs involved in the liver injury process?     

Answer: We have added more justification on why miR-29a is a good candidate to start with in the introduction section: “MicroRNA 29a has been identified as a critical molecule in drug-induced acute liver failure (1) inflammation (2), fibrosis (3-5), NAFLD(6), HCC (7), and metabolism (4). In addition, miR-29a participates in DNA methyltransferase regulation (8)”. Our co-author Dr. Kota has been conducting his research on miR29a and liver disease. However, no research has demonstrated the effect of HFD and HFD+ alcohol on miRNA expression, which is why we started with miR29a. Future analyses will focus on miRNA more broadly to enhance our mechanistic understanding.

  1. The authors should include experimental groups of animals fed with normal diets and EtOH alone for comparison.

Answer: Although we included a calorie-matched diet control recommended by the diet manufacturer, we thought it might make more sense to present the HFD and HFD + EtOH groups separately so as to avoid adding another variable to the study. The addition of CMD + EtOH groups was not performed,  as this does not directly address the question we sought to answer. Please see also the answer to reviewer 1, question 2. Additionally, EtOH effect on liver damage has been intensively studied in the alcohol-induced liver disease group.

  1. Are the amounts of food consumption different amongst groups of animals in this study? The authors should provide these data and discuss if they could have impact on the results of the study.

Answer.  In the introduction, we have added the information on why food intake was not measured. Prior research demonstrated that alcohol has minimal effect on food intake. Our study is quite involved due to the inclusion of many groups and long treatment terms. Still, we recognize that not measuring food intake is a potential limitation. This point has been added in the discussion.

  1. Why does miR-29a change in its expression over the course of the treatment? Which tissues/cell types are responsible for the process? The authors should include some in vitro experiments to support their proposed models.

Answer. We appreciate reviewer’s comments and have searched for more alcohol and miR-29a related papers to strengthen our understanding and discussion. Indeed, limited research has studied miR-29a in response to alcohol. In one study, rats were treated with Perilly alcohol, a plant derivative, and found to have increased expression of miR-29a (9). Another piece of indirect evidence suggests that miR-29a is among the most significant miRs increased in blood extracellular vesicles in an alcoholic steatohepatitis model (10). From our research, we found that induction of miR-29a is associated with liver damage and its induction at 13 weeks reflects the “defense” response to injury. However, due to the continuous insult from HFD and EtOH, the deleterious liver condition prevents further induction of miR-29a, which steadily decreases with the progression of liver injury. Detailed studies to understand the involvement of particular cell-types is not the focus of this study, but in vitro experiments will be considered for future studies.

  1. Statistical analyses are missing in many figures (Figure 2B, 8A, 8C, and 9A)

Answer: Statistical analyses have been performed and significant markers were added

References

  1. Xiang DD, Liu JT, Zhong ZB, Xiong Y, Kong HY, Yu HJ, et al. MicroRNA-29a-3p Prevents Drug-Induced Acute Liver Failure through Inflammation-Related Pyroptosis Inhibition. Curr Med Sci. 2023;43(3):456-68.
  2. Lin HY, Yang YL, Wang PW, Wang FS, Huang YH. The Emerging Role of MicroRNAs in NAFLD: Highlight of MicroRNA-29a in Modulating Oxidative Stress, Inflammation, and Beyond. Cells. 2020;9(4).
  3. Xu XY, Du Y, Liu X, Ren Y, Dong Y, Xu HY, et al. Targeting Follistatin like 1 ameliorates liver fibrosis induced by carbon tetrachloride through TGF-beta1-miR29a in mice. Cell Commun Signal. 2020;18(1):151.
  4. Dalgaard LT, Sorensen AE, Hardikar AA, Joglekar MV. The microRNA-29 family: role in metabolism and metabolic disease. Am J Physiol Cell Physiol. 2022;323(2):C367-C77.
  5. Jing F, Geng Y, Xu XY, Xu HY, Shi JS, Xu ZH. MicroRNA29a Reverts the Activated Hepatic Stellate Cells in the Regression of Hepatic Fibrosis through Regulation of ATPase H(+) Transporting V1 Subunit C1. Int J Mol Sci. 2019;20(4).
  6. Jampoka K, Muangpaisarn P, Khongnomnan K, Treeprasertsuk S, Tangkijvanich P, Payungporn S. Serum miR-29a and miR-122 as Potential Biomarkers for Non-Alcoholic Fatty Liver Disease (NAFLD). Microrna. 2018;7(3):215-22.
  7. Yang YL, Tsai MC, Chang YH, Wang CC, Chu PY, Lin HY, et al. MIR29A Impedes Metastatic Behaviors in Hepatocellular Carcinoma via Targeting LOX, LOXL2, and VEGFA. Int J Mol Sci. 2021;22(11).
  8. Yang YL, Kuo HC, Wang FS, Huang YH. MicroRNA-29a Disrupts DNMT3b to Ameliorate Diet-Induced Non-Alcoholic Steatohepatitis in Mice. Int J Mol Sci. 2019;20(6).
  9. Rajabi S, Najafipour H, Sheikholeslami M, Jafarinejad-Farsangi S, Beik A, Askaripour M, et al. Perillyl alcohol and quercetin modulate the expression of non-coding RNAs MIAT, H19, miR-29a, and miR-33a in pulmonary artery hypertension in rats. Noncoding RNA Res. 2022;7(1):27-33.

10.       Eguchi A, Lazaro RG, Wang J, Kim J, Povero D, Willliams B, et al. Extracellular vesicles released by hepatocytes from gastric infusion model of alcoholic liver disease contain a MicroRNA barcode that can be detected in blood. Hepatology. 2017;65(2):475-90.

Round 2

Reviewer 1 Report

The addition of the control group (CMB) made the study well done and presented. The discussion was also improved. I still found some spelling mistakes, carefully proofread the manuscript. 

Some spelling mistakes were detected. I find the conclusion in the abstract complicated to understand. Maybe make one simple but concise conclusion.

Author Response

We appreciate reviewer’s comments and language was double-checked.  The abstract was revised, and this version should be more easily to understand.

Reviewer 2 Report

The authors have submitted a revision of their manuscript previously reviewed by this reviewer. In this submission, the authors have added text throughout the manuscript and included additional analysis. While the manuscript is now longer, it remains overall not particularly readable or understandable.  Unfortunately, some grammatical errors make the manuscript difficult to read and understand.

Author Response

As the reviewer acknowledged, after adding the additional group and additional analysis, the paper is now longer. The paper’s length is due to the nature of its comprehensive design and multiple time points of data collection, and we are afraid that shortening it will not benefit the reader. We appreciate the reviewer’s comments, and the paper was read and approved by the authors.  This version should be easier to understand.

Reviewer 3 Report

I thank the authors for addressing all the questions and concerns that I have raised. I still think this study lacks insights into the biology of miR-29a in liver injury induced by high fat and alcohol. The study does not explain why miR-29a is dynamically altered during the injury process. Nevertheless, as the authors suggest, this question could be further explored in the future.

The quality of language is fine.

Author Response

We thank the reviewer for their comments.  Future studies focusing on the mechanism of miR-29a expression and EtOH treatment appear promising.